# Joint duration-cost-quality optimization model for complex product supply chains under contingency conditions

**Yunzhe Li, Peng Dong** *, **Weimin Ye**

Naval University of Engineering, Wuhan, China

* star_manchu0424@163.com, 0913041005@nue.edu.cn

**Data Availability Statement:** All relevant data are within the paper.

**Funding:** The author(s) received no specific funding for this work.

## Abstract

The Graphical Evaluation and Review Technique (GERT) and complex networks are used to simulate and analyse complex product supply chain networks based on the characteristics of complex product supply chain networks. And the traditional GERT is improved by constructing a grey parametric GERT network with restricted output results, taking into account the fact that the duration, product quality and product cost of each supplier in a complex product supply chain are interval values rather than definite values, and that customers have restrictions on the duration, product quality and product cost of the final product. The functional relationship between product quality, product cost and duration is analysed, and two satisfaction functions for duration and cost are constructed in order to quantify the multi-objective requirements of shortening duration, saving product cost and guaranteeing product quality for complex products under emergency situations. Then, a duration-cost-quality model for complex product supply chains in contingency situations is constructed to obtain the better duration, product cost and product quality of each supplier by optimising the indicator parameters in the network. Finally, the scientific validity and effectiveness of the model and method are verified by means of arithmetic example. The results show that the method is able to analyse the optimal duration, product quality and product cost of each supplier, and the main manufacturer can obtain an optimised combination of duration, cost and quality for a complex product supply chain in different contingency situation. To further promote the sustainable and secure development of complex product supply chains, this paper also suggests the integration of data sharing and blockchain technology with complex product supply chains to develop dynamic supply chain feedback management systems.

## 1. Introduction

When emergencies occur, the rapid and efficient response to the delivery needs of complex products required for the disposal of emergencies is of great importance to the timely and secure resolution of emergencies and the protection of national security and social development. Complex products are a category of large products or systems with complex customer requirements, complex product structures, and ultra-high added value, such as aerospace, large weapons and equipment, and large ships [1,2]. With economic globalisation, industrial

**Competing interests:** The authors have declared that no competing interests exist.

agglomeration and the continuous development of high technology, more and more manufacturers, such as Boeing and Airbus, are turning to the "main manufacturer-supplier" model, and a complex supply chain network is gradually formed through collaboration and innovation between main manufacturers and suppliers, with products as the link. A problem with any one supplier in the network can have a significant impact on the entire complex product supply chain, affecting delivery times, product quality and costs of complex products [3]. The Graphical Evaluation and Review Technique (GERT) is a network technology proposed by Pritsken in 1966 when researching the Apollo system's final launch time. The GERT network model can transform the complex system into a random network model with a simple structure through logical nodes and branches. It can not only give the random network diagram of the system but also does not need to analyze the given complex system in a recursive form. Project management tools, represented by critical path (CPM), Program Evalution and Review Technique (PERT), consider probabilistic uncertainty and are better suited to a wide range of areas such as analytical modelling, development management, quality management, reliability assessment and resource allocation. However, for the many non-probabilistic uncertainty problems that exist in reality, these management tools are no longer applicable. Compared with the CPM, PERT, the GERT network model has wider application fields, such as space research, cost optimization management [4], emergency plan management, project risk management [5], product development, manufacture and improvement plan, combat optimization problem [6], and computer algorithms for GERT networks [7]. The GERT network model has become a powerful tool for dealing classical probabilistic network relations [8]. Therefore, this paper (a) uses complex networks to simulate the construction of a complex product supply chain network structure, (b) explores the relationship between duration, product cost and product quality in a complex product supply chain network using duration-cost curves and duration-quality bell-shaped functions, (c) models the duration, quality and cost of complex product supply chains using an improved GERT technique, taking into account non-probabilistic uncertainty information, and (d) solves for optimal duration, optimal product quality and optimal product cost of each supplier in the supply chain network under different contingency conditions. Thereby, the article aims to achieve three objectives: (1) Meeting the requirements of shortening the duration, saving cost and guaranteeing quality of complex products under emergency situations, so as to respond to the delivery demand quickly and efficiently. (2) Improving and enriching the theoretical research related to the supply chain of complex products, especially the content related to the supply chain of complex products in emergency situations. (3) Providing decision basis for decision makers in the actual manufacturing of complex products based on the model findings.

This paper is organized sequentially (a) to analyse and describe complex product supply chain network structures using graphical evaluation and review technique, (b) to analyse the relationship between duration-cost and duration-quality of complex products, (c) to develop a joint duration-cost-quality optimization model for complex product supply chains under contingency conditions to solve problems, (d) to conduct a case study to verify and illustrate the model, and (e) to sum up conclusions and provide suggestions.

## 2. Literature review

1. Research on complex networks applied to supply chains. Complex network theory is an effective approach to the study of complex supply chains. A group of physicists, led by Dirk Helbing, have used complex network theory to conduct empirical research on relevant social networks. However, much of the research has concentrated on the nature and parameters of complex networks and has not really taken into account supply chain scenarios.

Moreover, the simple use of BA small-world networks (Proposed by Barabasi and Albert, also known as BA scale-free network) or ER scale-free networks (Proposed by Paul Erdős and Alfréd Rényi) for simulation makes the models lack a realistic basis and the conclusions obtained often do not reflect the reality of the situation [9]. Since the complexity of supply chain network structure, Yan et al. [10] presented a novel evaluation method to measure bullwhip effect in different network topologies and analysed the effect of network topologies and network size on the bullwhip effect. Zhu et al. [11] investigated the rules and characteristics of risk propagation in complex supply networks. Aiming at the characteristics of complex supply chain with complex structure and large risk consequences, considering the different impact capabilities and different anti-risk capabilities of each enterprise node, the author proposed a SITR (susceptible-infected-temporarily removed-completely removed) risk propagation model based on weighted networks and gave a supply chain risk control strategy. Wang and Zhang [12] defined upper and lower load limits for supply chain firms and found through simulation evolution that the upper limit was positively related to the size of cascading failures, while the lower limit was the opposite. Gao et al. [13] analysed complex manufacturing network systems in two dimensions, topological vulnerability and functional vulnerability, and gave recommendations for safe production. Kannan et al. [14] used a linear programming approach to analyse the impact of the physical location of suppliers located at the first layer of the supply chain. Zhang et al. [15] identified potential cascading failure problems in networks by studying the impact of initial events on the system causing maximum disruption consequences. Considering the cascading failure phenomenon as a dynamic optimisation problem, the author proposed an algorithm for identifying critical risks to analyse node failures. Sun et al. [16] developed a multi-echelon supply chain evolution model based on nodal degree preference and the connection preferences of competitive advantage coefficient, which provides a comprehensive assessment of supply chain robustness from multiple dimensions. As can be seen, many features of realistic supply chain networks have been reflected under the drilling of scholars and are summarised into various valid statistical parameters. At the same time scholars have improved complex network models by making them more consistent with realistic supply chain evolution. The application of game and cybernetic methods to modelling through disciplinary integration has further extended the applicability and validity of complex network theory.

2. Studies related to duration, quality and cost of complex products. The problem of complex products in the "main manufacturer-supplier" model has been studied by many scholars, focusing on two aspects: First, the research is carried out from the perspective of contract design based on game theory. Based on the evolutionary game, Xue et al. [17,18] proposed a dynamic penalty mechanism that is more conducive to complex product quality control from the perspective of contract design, and designed a contractual game model of quality improvement incentive gain-sharing contracts between the main manufacturers and suppliers of large ships, taking into account supply chain synergy factors. Liu and Wu [19] designed a class of quality incentive contracts for complex product supply chains by considering the effects of short-term economic benefits and long-term quality reputation, and by constructing a "benefit-reputation" utility function for quality cooperators in the supply chain to achieve a win-win situation in terms of their overall utility.

The second is a supply chain perspective based on the Graphical Evaluation and Review Technique (GERT). Wang et al. [20] proposed the concept of complex product quality value and established a GERT network model considering the value of complex product quality to improve the quality control capability and optimise the quality cost of complex product supply chain by identifying key suppliers in the supply chain. Based on the GERT network model, Bai

**Table 1. Overview of recent and relevant literatures.**

| Methodology | Location | Objective | Limitation |
|---|---|---|---|
| Game theory: (Evolutionary Game Theory, Principal-agent Theory, Stackelberg Game, etc.) | Complex products such as aerospace, large weaponry and large ships | Quality control of complex products, Collaborative incentive contracts for complex products | Most studies focus on the problems of one main manufacturer and one supplier, and fewer studies take a holistic view of the supply chain. |
| Graphical Evaluation and Review Technique | Complex products in the main manufacturer-supplier model | Quality management in the supply chain, Multi-objective joint optimisation of supply chains | Parameters assumptions in the model are mostly probabilistic uncertainties. |

[21] combined Bayesian methods, work structure decomposition, and particle swarm algorithms to study and simulate the accurate and effective control of weapons and equipment quality. Considering the uncertain factors in the construction process of projects, Han et al. [22] constructed a time-cost-quality joint optimization model based on interval GERT network to solve the problems of information feedback, rework iteration.

The literature shows that researches on complex product-related issues have produced some significant results, but it has also revealed the following problems: (a) In reality, a main manufacturer often works with multiple suppliers to form a network of supply chains, and many studies of complex products based on game theory from a contractual perspective are limited to one main manufacturer and one supplier, which has certain limitations. (b) When studying problems related to complex products based on GERT network models, most models have deterministic values for network parameters such as product quality, delivery time and product cost, which do not correspond to the actual situation where such parameters are non-deterministic. In some cases, although the non-deterministic nature of the network parameters is taken into account, the model is limited to the individual enterprise and is not applied to the supply chain network. (c) There is currently little exploration of complex product supply chains in emergency situations. Table 1 presents a general overview of recent and relevant literatures.

## 3. Structural characteristics of complex product supply chain networks

The supply chain network for complex products has the following characteristics:

1. **Complex supply chain network with many actors involved**
   Due to the complex structure of complex products, the large number of components and the many interested parties involved, the supplier network of the Airbus A380, for example, has 480 suppliers from 5 continents, making it a complex supply chain network.

2. **Close synergy and interaction between the various stakeholders**
   At the end of 2009, for example, Toyota faced an unprecedented crisis when tens of thousands of newly launched vehicles were urgently recalled from various countries due to brake failure, with heavy losses for the rest of the supply chain, both suppliers and sellers. In 2010, Boeing was testing the Boeing 787 passenger jet when a fire broke out, causing delays in the delivery of the aircraft [9].
   Graphical evaluation and review technique is a stochastic network simulation technique that can appropriately describe and reflect the characteristics and structure of complex networks, therefore this paper will be based on GERT networks to describe and reflect the network structure of complex product supply chains [23].

## 4. Problem description

The problem can be described as finding the optimal combination of schedule, cost and product quality of suppliers at each network node in a complex product supply chain network to meet the supply demand associated with a complex product in a contingency situation. The definitions and relevant assumptions of some specialized vocabularies involved in this study are as follows.

**(a) The concept of grey parameters and restricted output results**

Due to the presence of incomplete information and uncertainty, the parameter in the actual activity is often not an exact value, but an interval number. Therefore grey numbers are introduced to represent the parameter values [24].

**Definition 1.** A GERT network with a grey number of parameters is called a Grey GERT (denoted as G-GERT) network.

**Definition 2.** A duration parameter of interval number is called grey duration, a cost parameter of interval number is called grey cost and a quality parameter of interval number is called grey quality.

**Definition 3.** In actual supply chain operations, customers at the lower end of the supply chain often impose corresponding requirements on schedule, quality and cost, i.e. the output of a G-GERT network is restricted. Therefore, the definition of a G-GERT network with restricted output results is called RG-GERT network (Restricted Grey GERT, denoted as RG-GERT).

**(b) Basic building blocks and parameter representation of complex product supply chain networks**

The RG-GERT network consists of three elements: nodes, arrows and streams. The basic building blocks of the RG-GERT network are shown in Fig 1.

where node $i$ and node $j$ are the upstream and downstream companies in the supply chain. $U_{ij}$ denotes the flow from node $i$ to node $j$, reflecting the quantitative relationship between the passing activities between the nodes. $p_{ij}$ denotes the probability of the arrow line being realized when node $i$ is realized. $t(\otimes)_{ij}$ is the grey time required for the arrow line to occur. And $t(\otimes)_{ij} \in \left[\underline{t_{ij}}, \overline{t_{ij}}\right]$, $\underline{t_{ij}}$ and $\overline{t_{ij}}$ are the minimum and maximum values of time required to get from $i$ to $j$ respectively. $q(\otimes)_{ij}$ is the grey quality required for the arrow line to occur. And $q(\otimes)_{ij} \in \left[\underline{q_{ij}}, \overline{q_{ij}}\right]$, $\underline{q_{ij}}$ and $\overline{q_{ij}}$ are the minimum and maximum values of quality required to be achieved from $i$ to $j$ respectively. $c(\otimes)_{ij}$ is the grey cost required for the arrow line to occur. And $c(\otimes)_{ij} \in \left[\underline{c_{ij}}, \overline{c_{ij}}\right]$, $\underline{c_{ij}}$ and $\overline{c_{ij}}$ are the minimum and maximum values of cost required to be spent from $i$ to $j$ respectively. $T^{RG}$ indicates the duration limit of the output result, $Q^{RG}$ is the quality required for the output result and $C^{RG}$ is the cost limit for the output result.

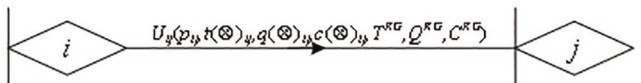

**Fig 1. Diagram of the basic RG-GERT network components.**

**(c) Moment Generating Function, transfer functions and numerical characteristics of parameters**

**Definition 4.** Assume that the probability of the covariate $x_{ij}$ is $p_{ij}$, and the probability density function of the covariate $x_{ij}$ for activity $(i, j)$ is $f(x_{ij})$. The moment generating function $M_{x_{ij}}(s)$ and the transfer function $W_{x_{ij}}(s)$ of $x_{ij}$ are defined as follows:

$$M_{x_{ij}}(s) = \int_{-\infty}^{+\infty} e^{sx_{ij}} f(x_{ij}) \, dx_{ij} \tag{1}$$

$$W_{x_{ij}}(s) = p_{ij} \cdot M_{x_{ij}}(s) \tag{2}$$

**Lemma 1.** Assuming that the probability density function of a continuous random variable $X$ is $\begin{cases} f_X(x), & a < x < b \\ 0, & else \end{cases}$. $Y$ is a random variable, and $Y = g(X)$. $g(X)$ is a continuous function on the interval [a, b]. The inverse function $x = h(y)$ of $Y = g(X)$ is continuously derivable over the domain of definition $\Omega$. The probability density function $f_Y(y)$ of $Y$ is therefore as follows:

$$f_Y(y) = \left\{ \begin{array}{ll} \sum_i f_X[h_i(y)] \left| \frac{dh_i(y)}{dy} \right|, & y \in \Omega \\ 0, & else \end{array} \right\} \tag{3}$$

**Proof.** The interval $\Omega$ can be divided into many non-overlapping subintervals $\Omega_i$, and $\Omega_i = [a_{(i)}, b_{(i)}]$. $F_Y(y)$ is the distribution function of $Y$, and $F_Y(y) = P\{Y \leq y\} = P\{g(X) \leq y\}$.

1. When $g'(X) > 0$, then $\frac{dh_i(y)}{dy} > 0$.

   $F_Y(y) = \sum_i P\{x_i \leq h_i(y)\}$, then $F_Y(y) = F_y'(y) = \sum_i f_X[h_i(y)] \frac{dh_i(y)}{dy} = \sum_i f_X[h_i(y)] \left| \frac{dh_i(y)}{dy} \right|$.

2. When $g'(X) < 0$, then $\frac{dh_i(y)}{dy} < 0$.

   $F_Y(y) = \sum_i P\{x_i \geq h_i(y)\} = 1 - \sum_i P\{x_i \leq h_i(y)\}$, then
   $f_Y(y) = F_Y'(y) = -\sum_i f_X[h_i(y)] \frac{dh_i(y)}{dy} = \sum_i f_X[h_i(y)] \left| \frac{dh_i(y)}{dy} \right|$.

3. When $g_j'(X) > 0, g_k'(X) < 0$, then $\frac{dh_j(y)}{dy} > 0, \frac{dh_k(y)}{dy} < 0$.

   $F_Y(y) = F_Y y = \sum_j P\left\{x_j \leq h_j(y)\right\} + \sum_k P\{x_k \geq h_k(y)\}$, then
   $f_Y(y) = F_Y'y = \sum_j f_X\left[h_j(y)\right] \frac{dh_j(y)}{dy} - \sum_k f_X[h_k(y)] \frac{dh_k(y)}{dy} = \sum_i f_X[h_i(y)] \left| \frac{dh_i(y)}{dy} \right|$.

**Lemma 2.** In RG-GERT networks, assuming that the probability density function of a continuous random variable $X$ is $f_X(x)$. $Y$ is a random variable, and $Y = g(X)$, $x = h(y)$ is the inverse function of $Y$, then the moment generating function $M_{x_{ij}}(s)$ of $Y$ is as follows:

$$M_Y(s) = \int_{-\infty}^{+\infty} e^{sg(X)} f_X[h(y)] dx \tag{4}$$

**Proof.**
$M_Y(s) = \int_{-\infty}^{+\infty} e^{sy} f_Y(y) dy = \int_{-\infty}^{+\infty} e^{sg(x)} f_X[h(y)] \frac{dh(y)}{dy} dy = \int_{-\infty}^{+\infty} e^{sg(x)} f_X[h(y)] \frac{dh(y)}{dy} \frac{1}{\frac{dh(y)}{dy}} dx$, therefore $M_Y(s) = \int_{-\infty}^{+\infty} e^{sg(x)} f_X[h(y)] dx$.

**Lemma 3.** In RG-GERT networks, assuming that the parameters $t(\otimes)_{ij}$, $q(\otimes)_{ij}$, $c(\otimes)_{ij}$ in the activity $(i, j)$ are independent of each other, and the equivalence parameter $X_{Eij} = t_{ij} + q_{ij} + c_{ij}$

Then, the equivalence moment generating function $M_E(s)$ is as follows:

$$M_E(s) = M_{t_{ij}}(s_1) \cdot M_{q_{ij}}(s_2) \cdot M_{c_{ij}}(s_3) \tag{5}$$

**Proof.**

$$M_E(s) = E[e^{sx_{Eij}}] = E\left[e^{\left(s_1 t_{ij} + s_2 q_{ij} + s_3 c_{ij}\right)}\right] = E[e^{s_1 t_{ij}}] \cdot E[e^{s_2 q_{ij}}] \cdot E[e^{s_3 c_{ij}}] = M_{t_{ij}}(s_1) \cdot M_{q_{ij}}(s_2) \cdot M_{c_{ij}}(s_3).$$

The parameters of activity $(i, j)$ in the RG-GERT network are $t_{ij}, q_{ij}, c_{ij}$, corresponding to three dimensions $s_1, s_2, s_3$, respectively, with relations satisfying $q_{ij} = q_{ij}(t_{ij})$, $c_{ij} = c_{ij}(t_{ij})$. Combining Lemma 1, Lemma 2 and Lemma 3 the following corollaries can be deduced.

**Corollary 1.** The equivalence moment generating function $M(s_1, s_2, s_3)$ in the activity $(i, j)$ is as follows:

$$M(s_1, s_2, s_3) = M_t(s_1) \cdot M_{q(t)}(s_2) \cdot M_{c(t)}(s_3) \tag{6}$$

**Corollary 2.** Assuming that $W_E(s_1, s_2, s_3)$ is the equivalent transfer function in the RG-GERT networks, then the equivalent probability $p_E$ and the equivalent moment generating function $M_E(s)$ are as follows:

$$p_E = W_E(s_1, s_2, s_3)|_{s_1 = s_2 = s_3 = 0} = W_E(0, 0, 0) \tag{7}$$

$$M_E(s) = \frac{W_E(s_1, s_2, s_3)}{W_E(0, 0, 0)} \tag{8}$$

**Proof.** From definition 4, $W_E(s_1, s_2, s_3) = p_E \cdot M_{t_{ij}}(s_1) \cdot M_{q_{ij}}(s_2) \cdot M_{c_{ij}}(s_3)$. When $s_1 = s_2 = s_3 = 0$, $M_{t_{ij}}(0) = M_{q_{ij}}(0) = M_{c_{ij}}(0) = 1$. Therefore, $M_E(s) = \frac{W_E(s_1, s_2, s_3)}{W_E(0,0,0)}$, $p_E = W_E(s_1, s_2, s_3)|_{s_1 = s_2 = s_3 = 0} = W_E(0, 0, 0)$.

**Corollary 3.** The expectations for each parameter are as follows:

$$E(t) = \frac{\partial}{\partial s_1}\left[\frac{W_E(s_1, s_2, s_3)}{W_E(0, 0, 0)}\right]\Bigg|_{s_1 = s_2 = s_3 = 0} \tag{9}$$

$$E(q) = \frac{\partial}{\partial s_2}\left[\frac{W_E(s_1, s_2, s_3)}{W_E(0, 0, 0)}\right]\Bigg|_{s_1 = s_2 = s_3 = 0} \tag{10}$$

$$E(c) = \frac{\partial}{\partial s_3}\left[\frac{W_E(s_1, s_2, s_3)}{W_E(0, 0, 0)}\right]\Bigg|_{s_1 = s_2 = s_3 = 0} \tag{11}$$

**(d) Equivalent transfer function calculation**

In RG-GERT networks, the equivalent transfer function $W_E(s_1, s_2, s_3)$ from node $i$ to node $j$ can be calculated according to the Mason's Formula as follows:

$$W_E(s_1, s_2, s_3) = \frac{\sum_{k=1}^{2} H_Z \cdot W_Z}{H} \tag{12}$$

Where $H$ is the characteristic equation. $H = 1 - \Sigma$(transmission coefficient of the odd-order loop) $+ \Sigma$(transmission coefficient of the even-order loop). $H_Z$ denotes the characteristic equation after the elimination of all the nodes and arrows that are associated with the $z$-th path. $W_Z$ is the equivalent transfer function on the $z$-th path from node $i$ to node $j$.

In summary, the principles of modelling complex product RG-GERT networks and the steps for their analysis can be seen as follows:

**Step 1.** Analyse the basic characteristics of a complex product supply chain and use them to construct the RG-GERT network model.

**Step 2.** Collect the basic parameters of each activity in the network and obtain the transfer function $W_{ij}$ by means of the probability $p_{ij}$ and the moment generating function $M_{ij}(s)$.

**Step 3.** Apply the Mason's formula to solve for the equivalent transfer function $W_E(s)$ of the network.

**Step 4.** Solve for the equivalence probability $p_E$, the equivalence moment generating function $M_E(s)$ and the basic parameters in the network according to the definition of the equivalence transfer function $W_E(s)$ and the properties of the moment generating function.

**e) Complex product cost-duration relationship**

The costs of complex products usually include direct and indirect costs. Direct costs include labour and material costs, etc. The shorter the duration of a complex product, the higher the direct costs. Indirect costs include management and other costs, and the longer the production period of a complex product, the higher the indirect costs. The relationship between total cost and duration is generally parabolic [22], as shown in Fig 2.

This paper assumes that the cost and duration of production of a complex product satisfy the equation: $c_{ij} = k_{ij}\left(t_{ij} - t^*_{ij}\right)^2 + \underline{c_{ij}}$. Where $c_{ij}$, $t_{ij}$ and $t^*_{ij}$ are the cost, duration and optimum duration of the process for activity $(I, j)$ respectively. Costs are lowest at $t_{ij} = t^*_{ij}$. $k_{ij}$ indicates how quickly costs vary with duration and $k_{ij} > 0$, $t^*_{ij} = \alpha_1 \underline{t_{ij}} + \alpha_2 \overline{t_{ij}}$, $\alpha_1 + \alpha_2 = 1$. When $\alpha_1 > 0.5$, $c_{ij}$ is the maximum at $t_{ij} = \overline{t_{ij}}$. when $\alpha_1 > 0.5$, $c_{ij}$ is the maximum at $t_{ij} = \overline{t_{ij}}$, and when $\alpha_1 > 0.5$, $c_{ij}$ is the maximum at $t_{ij} = \overline{t_{ij}}$ or $t_{ij} = \underline{t_{ij}}$.

**f) Complex product quality-duration relationship**

For complex products, too long or too short a duration is not conducive to achieving the highest quality, and the relationship between duration and product quality for complex products is usually bell-shaped [22], as shown in Fig 3.

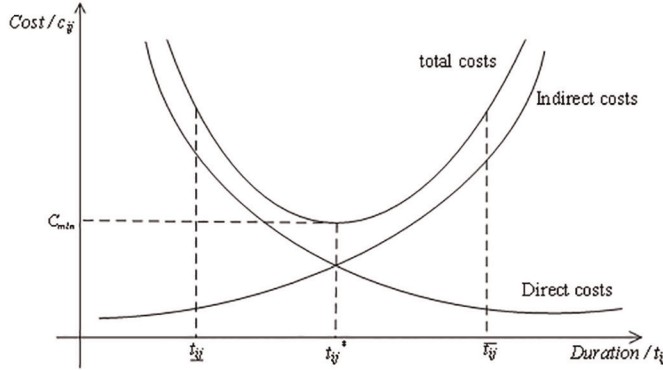

**Fig 2. Cost-duration relationship.**

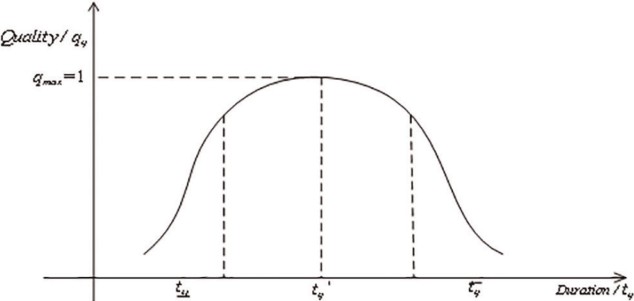

**Fig 3. Quality-duration relationship.**

This paper assumes that complex product quality and duration satisfy the equation:
$q_{ij} = 1/1 + \left| \frac{t_{ij} - t^*_{ij}}{\overline{t_{ij}} - \underline{t_{ij}}} \right|^{k_t}$. Where $t^*_{ij} = \beta_1 \underline{t_{ij}} + \beta_2 \overline{t_{ij}}$, $k_t$ indicates the width and flatness of the bell-shaped function and $k_t > 0$, $\beta_1 + \beta_2 = 1$. Product quality is highest at $t_{ij} = t^*_{ij}$.

**g) Duration satisfaction function**

**Definition 5.** For the duration parameters of complex products, the duration satisfaction function $\mu_1$ is defined as follows.

$$\mu_1 = \begin{cases} 1, & E(t) \leq T^{RG} \\ \frac{\delta_1 - E(t)}{\delta_1 - T^{RG}}, & T^{RG} \leq E(t) \leq \delta_1 \\ 0, & else \end{cases} \tag{13}$$

Where $\delta_0$ indicates that complex products cannot be built earlier than $\delta_0$, otherwise problems such as quality may arise. $\delta_1$ indicates that complex products cannot be built later than $\delta_1$, otherwise it may interfere with the use of the product or delay the product from capturing the market. $T^{RG}$ is the customer's expectation of the duration required to produce a complex product. And when complex products are out of range $[\delta_0, \delta_1]$, customer satisfaction is 0, as shown in Fig 4.

**h) Cost satisfaction function**

**Definition 6.** For the cost parameters of complex products, the cost satisfaction function $\mu_2$ is defined as follows.

$$\mu_2 = \begin{cases} 1, & E(c) \leq \delta_2 \\ \frac{\delta_3 - E(c)}{\delta_3 - \delta_2}, & \delta_2 \leq E(c) \leq \delta_3 \\ 0, & else \end{cases} \tag{14}$$

Where $[\delta_2, \delta_3]$ indicates the range of fluctuations in the cost of complex products that customers can tolerate. Customer satisfaction is 0 when the cost of a complex product exceeds $\delta_3$. Customer satisfaction is 1 when the cost of a complex product does not exceed $\delta_2$, as shown in Fig 5.

## 5. Construction and solution of a duration-cost-quality model for complex product supply chains in contingency situations

**(1) Construction of a duration-cost-quality model**

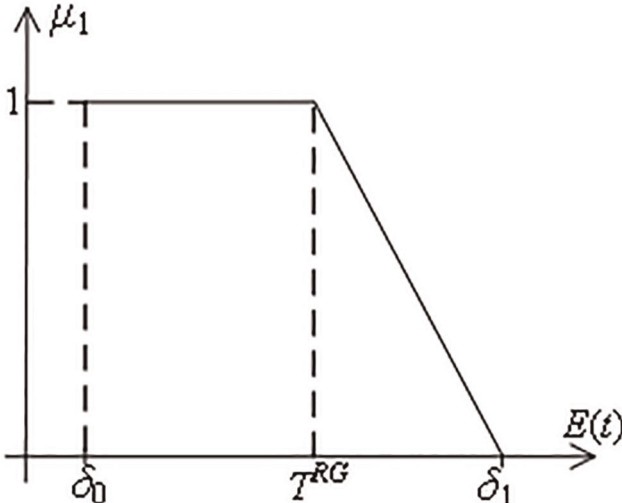

**Fig 4. Function of the duration satisfaction $\mu_1$.**

The RG-GERT network is used to simulate a complex product supply chain network, and the RG-GERT network is solved to obtain the duration $E(t)$, cost $E(c)$ and product quality $E(q)$ of the complex product. Since multi-objective optimization problems are difficult to find optimal solutions, this paper converts the multi-objective optimization problem into a single-objective optimization problem with customer requirements, product characteristics as constraints and maximum customer satisfaction as the objective, and constructs a duration-cost-quality model

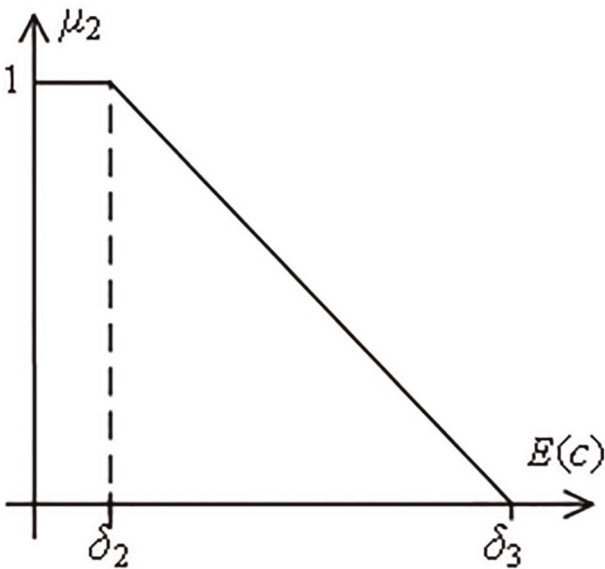

**Fig 5. Function of the cost satisfaction $\mu_2$.**

for complex products in contingency situations. The model is as follows.

$$
s.t
\begin{cases}
\quad max\ \mu \\[4pt]
\mu = \omega_1 \mu_1 + \omega_2 \mu_2 \\[4pt]
\mu_1 = \begin{cases}
1, & \delta_0 \le E(t) \le T^{RG} \\[6pt]
\dfrac{\delta_1 - E(t)}{\delta_1 - T^{RG}}, & T^{RG} \le E(t) \le \delta_1 \\[6pt]
0, & else
\end{cases} \\[30pt]
\mu_2 = \begin{cases}
1, & E(c) \le \delta_2 \\[6pt]
\dfrac{\delta_3 - E(c)}{\delta_3 - \delta_2}, & \delta_2 \le E(c) \le \delta_3 \\[6pt]
0, & else
\end{cases} \\[30pt]
\underline{t_{ij}} \le t_{ij} \le \overline{t_{ij}} \\[6pt]
\underline{q_{ij}} \le q_{ij} \le \overline{q_{ij}} \\[6pt]
\underline{c_{ij}} \le c_{ij} \le \overline{c_{ij}} \\[6pt]
\delta_0 \le E(t) \le \delta_1 \\[6pt]
\delta_5 \le E(q) \le \overline{Q} \\[6pt]
\underline{c} \le E(c) \le \overline{c} \\[6pt]
q_{ij} = 1/1 + \left| \dfrac{t_{ij} - t^*_{ij}}{\overline{t_{ij}} - \underline{t_{ij}}} \right|^{k_t} \\[10pt]
c_{ij} = k_{ij} \left( t_{ij} - t^*_{ij} \right)^2 + \underline{c_{ij}} \\[6pt]
\omega_1 + \omega_2 = 1
\end{cases}
$$

Where $\mu$ is total customer satisfaction. $E(t)$, $E(q)$ and $E(c)$ can be found according to Corollary 3. $\omega_1$ is the weight of duration satisfaction for complex products and $\omega_2$ is the weight of cost satisfaction. $\omega_1$ and $\omega_2$ can be obtained by Delphi, AHP hierarchical analysis, customer designation, etc.

### (2) Solution of a duration-cost-quality model

Complex product supply chain duration-cost-quality joint optimisation model problems are non-linear optimisation problems. When solving such problems, classical non-linear programming algorithms have strong local search capability, but weak global search capability and strong dependence on initial values. Intelligent algorithms have a greater advantage over classical non-linear programming algorithms in terms of global search capability. Considering the problem of solution efficiency, this paper uses the particle swarm algorithm [25,26] to solve the problem, and introduces the probability of variation in the particle swarm position update to improve the global search capability of the algorithm. The main solution process and steps are shown in Fig 6.

Solution steps:

**Step 1.** Initialise the population parameters and positions. These include: population size, dimensionality (number of variables), maximum number of iterations, self-learning factor, group learning factor, inertia weights, maximum velocity, velocity and position of the initial population.

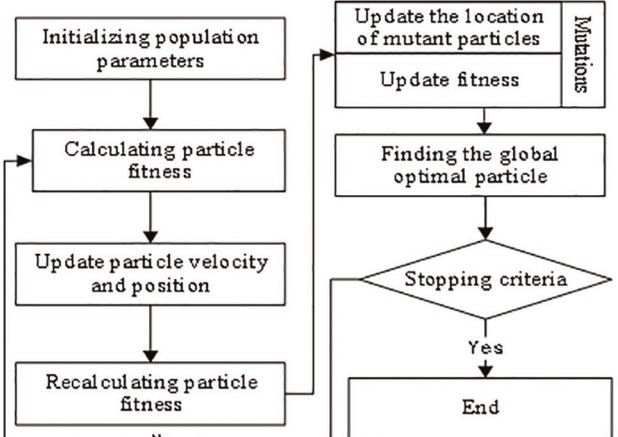

**Fig 6. Figure model solving process.**

**Step 2.** Calculate the fitness of the particles. The main components are: fitness calculation for the population, determination of individual historical best position and historical best fitness, population historical best position and population historical best fitness.

**Step 3.** Update the velocity and position of the particle according to the velocity and position update formulae and the boundary is processed. As particle n in generation t evolves towards generation (t+1), its velocity and position are updated by the following equations:

$$v_n(t+1) = \omega \cdot v_n(t) + c_1 \cdot r_1(t) \cdot [p_n(t) - x_n(t)] + c_2 \cdot r_2(t) \cdot \left[p_g(t) - x_n(t)\right] \qquad (15)$$

$$x_n(t+1) = x_n(t) + v_n(t+1) \qquad (16)$$

where $x_n(t)$ is the current position of particle $n$ in generation $t$. $p_n(t)$ is the individual historical best position in generation $t$, and $p_g(t)$ is the population historical best position in generation $t$. $v_n(t)$ is the current velocity of particle $n$ in generation $t$. $\omega$ is the inertia weight, $c_1$ is the self-learning factor, and $c_2$ is the population learning factor.

**Step 4.** Particle mutation. The mutation probability is introduced, and when the particle mutates, the particle $n$ in generation (t+1) is randomly selected for position update and the boundary is processed. The mutation update formula is as follows:

$$x_n(t+1) = x_{n\_max} + (x_{n\_max} - x_{n\_min}) \cdot rand \qquad (17)$$

where $x_{n\_max}$ is the maximum position point of particle $n$, $x_{n\_min}$ is the minimum position point of particle $n$, and rand is a random number between 0 and 1.

**Step 5.** The fitness of the updated and mutated particles is calculated, and the new fitness is compared with the individual historical best fitness to update the individual historical best fitness, and then the individual historical best fitness is compared with the population historical best fitness to find the globally optimal particle.

**Step 6.** If the stopping criteria is reached, then the process ends. Otherwise, go back to step 2.

## 6. Case study

The production of large aviation products is a typical application of the "main manufacturer-supplier" model, which is a typical representative of complex products. In this production model, the manufacturing mechanism of complex products is expressed as follows: the customer proposes requirements to the main manufacturer, who decomposes the target according to the customer requirements to form the requirements of the product for the lower layers of suppliers. Lower layers of suppliers, based on their product requirements, will again decompose the requirements to form the objectives for their lower layers of suppliers. This is broken down in a similar manner to form product objectives for each layer of suppliers. The lower layer suppliers deliver the products to the upper layer suppliers after completing the production of the components according to the product requirements. By analogy, the main manufacturer completes the final assembly and inspection of the product to ensure that the product meets the customer's requirements, and then delivers the product to the customer, the realisation mechanism is shown in Fig 7.

This paper constructs an RG-GERT network based on the production of a supporting product for a large civil aircraft. For the sake of research and not generality, this paper assumes that the supplier network consists of one customer, one main manufacturer and three suppliers,

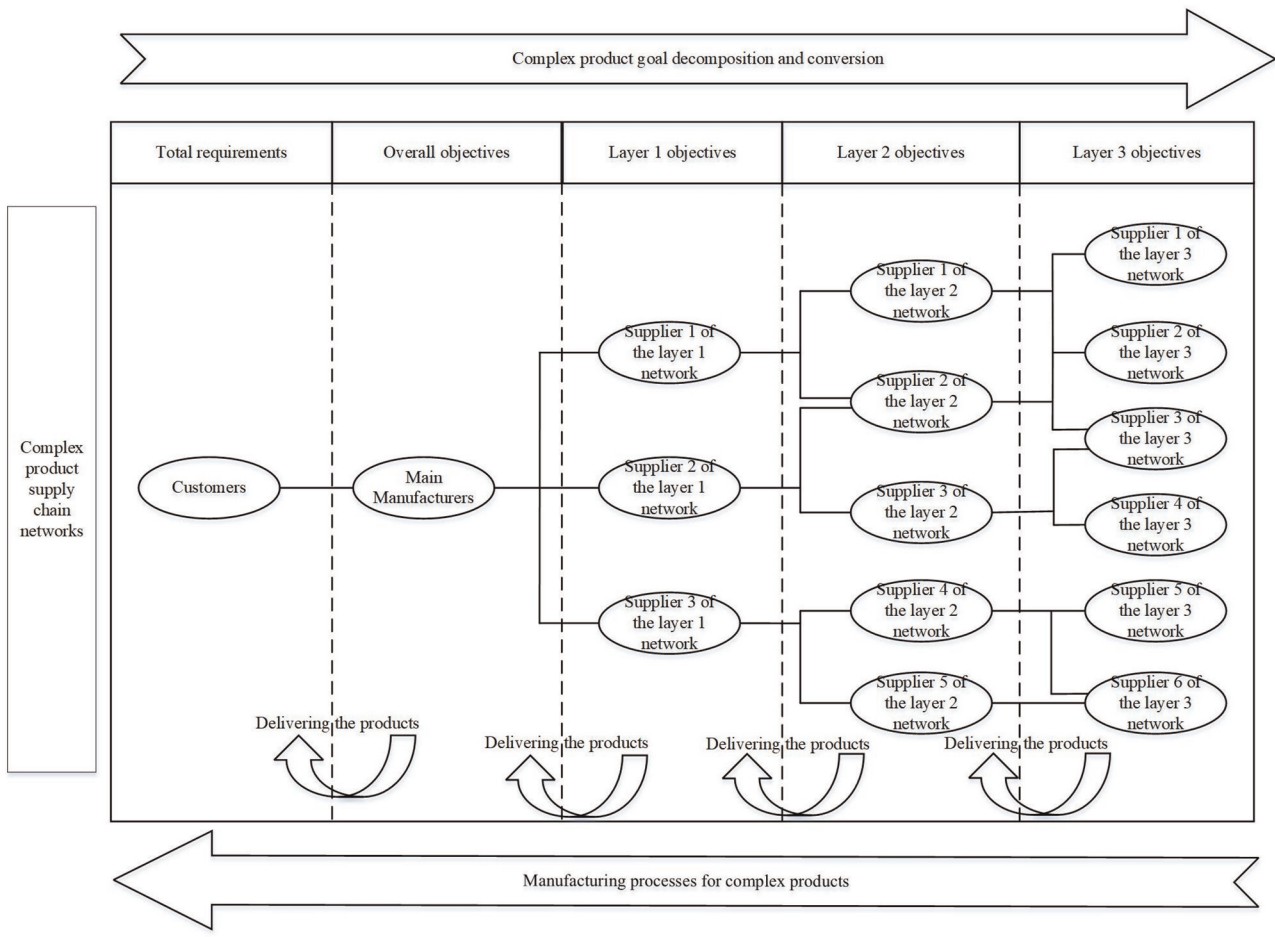

**Fig 7. The production processes and mechanisms of complex products in the supply chain.**

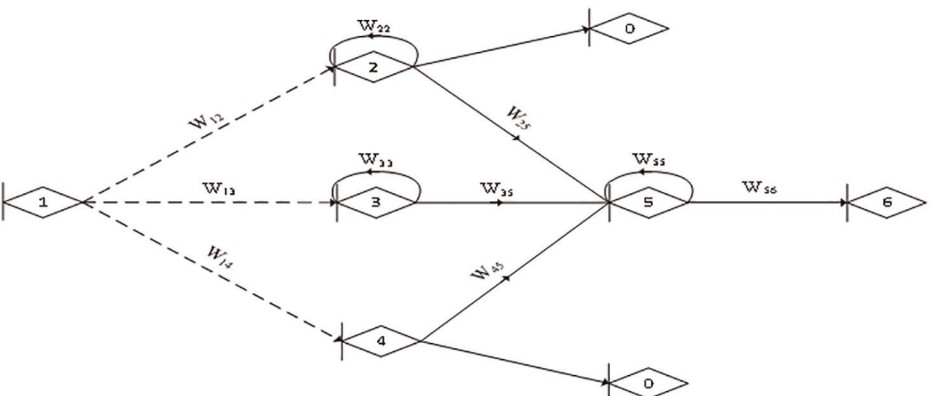

**Fig 8. The RG-GERT network model of a supporting product for a large civil aircraft supply chain.**

and Fig 8 depicts its RG-GERT network model and the specifics of each activity process in the supply network are as shown in the Table 2.

Where node 1 represents the collective name of the layer 2 suppliers in the supply chain network. Nodes 2, 3, and 4 represent that there are three layer 1 suppliers in the supply chain network. Node 5 represents the main manufacturer in this supply chain network and node 6 represents the customers in this supply chain network.

- **Explanation of the process for each activity:** Table 2 explains the specifics of each activity process in the supply network.

- **Network analysis:** Fig 8 shows that the network consists of three 1st order loops, three 2nd order loops and one 3rd order loop, as shown in Table 3.

Therefore, based on Eq (12), the characteristic equation $H$: $H = 1 - W_{22} - W_{33} - W_{55} + W_{22} \cdot W_{33} + W_{22} \cdot W_{55} + W_{33} \cdot W_{55} - W_{22} \cdot W_{33} \cdot W_{55}$.

**Table 2. Process of activities.**

| Activities | The Process |
|---|---|
| (1, 2) | Supply of products from suppliers of layer 2 to suppliers 2 of layer 1 |
| (1, 3) | Supply of products from suppliers of layer 2 to suppliers 3 of layer 1 |
| (1, 4) | Supply of products from suppliers of layer 2 to suppliers 4 of layer 1 |
| (2, 0) | Supplier 2 Scrapping of manufactured products |
| (2, 2) | Supplier 2 supplied the main manufacturer 5 with products that failed the inspection and the products were reworked. |
| (2, 5) | Supplier 2 supplies products to the main manufacturer 5 |
| (3, 3) | Supplier 3 supplied the main manufacturer 5 with products that failed the inspection and the products were reworked |
| (3, 5) | Supplier 3 supplies products to the main manufacturer 5 |
| (4, 0) | Supplier 4 Scrapping of manufactured products |
| (4, 5) | Supplier 4 supplies products to the main manufacturer 5 |
| (5, 5) | The main manufacturer 5 supplied the customer with a product that failed the inspection and the product was reworked |
| (5, 6) | The main manufacturer 5 supplies products to the customer |

**Table 3. Parameter of Mason Formula for the RG-GERT network.**

| 1-order loop | Transfer functions | 2-order loop | Transfer functions | 3-order loop | Transfer functions |
|---|---|---|---|---|---|
| L1: 2→2 | $W_{22}$ | L4: 2→2, 3→3 | $W_{22} \cdot W_{33}$ | L7: 2→2, 3→3, 5→5 | $W_{22} \cdot W_{33} \cdot W_{55}$ |
| L2: 3→3 | $W_{33}$ | L5: 2→2, 5→5 | $W_{22} \cdot W_{55}$ | | |
| L3: 5→5 | $W_{55}$ | L6: 3→3, 5→5 | $W_{33} \cdot W_{55}$ | | |

Path 1→2→ 5→6: $H_1 = 1\text{-}W_{33}$, $W_1 = W_{12} \cdot W_{25} \cdot W_{56}$. Path 1→3→5→6: $H_2 = 1\text{-}W_{22}$, $W_2 = W_{13} \cdot W_{35} \cdot W_{56}$. Path 1→4→5→6: $H_3 = 1\text{-}W_{22}\text{-}W_{33}+W_{22} \cdot W_{33}$, $W_3 = W_{14} \cdot W_{45} \cdot W_{56}$.

The equivalent transfer function $W_E(s)$ of the network is as follows:

$$W_E(s) = \frac{\sum_{k+1}^{3} H_K \cdot W_K}{H}$$

$$= \frac{W_{12} \cdot W_{25} \cdot W_{56} \cdot (1 - W_{33}) + W_{13} \cdot W_{35} \cdot W_{56} \cdot (1 - W_{22}) + W_{14} \cdot W_{45} \cdot W_{56} \cdot (1 - W_{22} - W_{33} + W_{22} \cdot W_{33})}{1 - W_{22} - W_{33} - W_{55} + W_{22} \cdot W_{33} + W_{22} \cdot W_{55} + W_{33} \cdot W_{55} - W_{22} \cdot W_{33} \cdot W_{55}}.$$

### (1) Under normal conditions

Depending on the characteristics of the product, it may have more serious problems if the duration is less than 66 days, and more than 86 days will delay the use of the product. In terms of cost, the acceptable range for the customer is 450 thousand RMB to 640 thousand RMB. In terms of quality, a relative quality control ranging from 0.9 to 1 is required. From Fig 1, we know that $t(\otimes)_{ij}$ is the grey time required for the arrow line to occur. $\underline{t_{ij}}$ and $\overline{t_{ij}}$ are the minimum and maximum values of time required to get from $i$ to $j$ respectively and $t(\otimes)_{ij} \in \left[\underline{t_{ij}}, \overline{t_{ij}}\right]$. $q(\otimes)_{ij}$ is the grey quality required for the arrow line to occur. And $q(\otimes)_{ij} \in \left[\underline{q_{ij}}, \overline{q_{ij}}\right]$, $\underline{q_{ij}}$ and $\overline{q_{ij}}$ are the minimum and maximum values of quality required to be achieved from $i$ to $j$ respectively. $c(\otimes)_{ij}$ is the grey cost required for the arrow line to occur. And $c(\otimes)_{ij} \in \left[\underline{c_{ij}}, \overline{c_{ij}}\right]$, $\underline{c_{ij}}$ and $\overline{c_{ij}}$ are the minimum and maximum values of cost required to be spent from $i$ to $j$ respectively. The parameters follow a normal distribution and $\delta_t, \delta_c, \delta_q$ are standard deviations. The RG-GERT network activity parameters are shown in Table 4, based on statistical data and normalised.

Therefore, based on the Eqs (7)-(11), the equivalence transfer probability $P_E$, duration expectation $E(t)$, quality expectation $E(q)$ and cost expectation $E(c)$ are as follows:

$$P_E = 0.9636;$$

$$E(t) = 0.3223*t_{12} + 0.3425*t_{13} + 0.3352*t_{14} + 0.05688*t_{22} +$$
$$0.3223*t_{25} + 0.06044*t_{33} + 0.3425*t_{35} + 0.3352*t_{45} + 0.06383*t_{55} + t_{56};$$

$$E(c) = 0.3223*c_{12} + 0.3425*c_{13} + 0.3352*c_{14} + 0.05688*c_{22} +$$
$$0.3223*c_{25} + 0.06044*c_{33} + 0.3425*c_{35} + 0.3352*c_{45} + 0.06383*c_{55} + c_{56};$$

$$E(q) = 0.1013*q_{12} + 0.1077*q_{13} + 0.1054*q_{14} + 0.01788*q_{22} +$$
$$0.1013*q_{25} + 0.019*q_{33} + 0.1077*q_{35} + 0.1054*q_{45} + 0.02006*q_{55} + 0.3143*q_{56};$$

**Table 4. Parameter of the RG-GERT network activity.**

| Activities | Probability | Duration / day | Cost / ten thousand | Quality / 1 | Distribution |
|:---:|:---:|:---:|:---:|:---:|:---:|
| (1, 2) | 0.33 | $t(\otimes)_{12}\epsilon[5,10]$ $\delta_{t12}{}^2 = 1$ | $c(\otimes)_{12}\epsilon[4,8]$ $\delta_{c12}{}^2 = 0.8$ | $q(\otimes)_{12}\epsilon[0.9,1]$ $\delta_{q12}{}^2 = 0.02$ | Normal |
| (1, 3) | 0.33 | $t(\otimes)_{13}\epsilon[5,10]$ $\delta_{t13}{}^2 = 1$ | $c(\otimes)_{13}\epsilon[4,8]$ $\delta_{c13}{}^2 = 0.8$ | $q(\otimes)_{13}\epsilon[0.9,1]$ $\delta_{q13}{}^2 = 0.02$ | Normal |
| (1, 4) | 0.34 | $t(\otimes)_{14}\epsilon[5,10]$ $\delta_{t14}{}^2 = 1$ | $c(\otimes)_{14}\epsilon[4,8]$ $\delta_{c14}{}^2 = 0.8$ | $q(\otimes)_{14}\epsilon[0.9,1]$ $\delta_{q14}{}^2 = 0.02$ | Normal |
| (2, 0) | 0.05 | $t(\otimes)_{20}\epsilon[7,30]$ | $c(\otimes)_{20}\epsilon[6,22]$ | $<0.85$ | —— |
| (2, 2) | 0.15 | $t(\otimes)_{22}\epsilon[7,9]$ $\delta_{t22}{}^2 = 0.4$ | $c(\otimes)_{22}\epsilon[6,7]$ $\delta_{c22}{}^2 = 0.2$ | $q(\otimes)_{22}\epsilon[0.85,1]$ $\delta_{q22}{}^2 = 0.03$ | Normal |
| (2, 5) | 0.8 | $t(\otimes)_{25}\epsilon[25,30]$ $\delta_{t25}{}^2 = 1$ | $c(\otimes)_{25}\epsilon[15,20]$ $\delta_{c22}{}^2 = 1$ | $q(\otimes)_{25}\epsilon[0.9,1]$ $\delta_{q25}{}^2 = 0.02$ | Normal |
| (3, 3) | 0.15 | $t(\otimes)_{33}\epsilon[4,8]$ $\delta_{t33}{}^2 = 0.8$ | $c(\otimes)_{33}\epsilon[3,6]$ $\delta_{c33}{}^2 = 0.6$ | $q(\otimes)_{33}\epsilon[0.85,1]$ $\delta_{q33}{}^2 = 0.03$ | Normal |
| (3, 5) | 0.85 | $t(\otimes)_{35}\epsilon[22,32]$ $\delta_{t35}{}^2 = 2$ | $c(\otimes)_{35}\epsilon[12,22]$ $\delta_{c35}{}^2 = 2$ | $q(\otimes)_{35}\epsilon[0.9,1]$ $\delta_{q35}{}^2 = 0.02$ | Normal |
| (4, 0) | 0.05 | $t(\otimes)_{40}\epsilon[26,29]$ | $c(\otimes)_{40}\epsilon[16,19]$ | $<0.9$ | —— |
| (4, 5) | 0.95 | $t(\otimes)_{45}\epsilon[26,29]$ $\delta_{t45}{}^2 = 0.6$ | $c(\otimes)_{45}\epsilon[16,19]$ $\delta_{c45}{}^2 = 0.6$ | $q(\otimes)_{45}\epsilon[0.9,1]$ $\delta_{q45}{}^2 = 0.02$ | Normal |
| (5, 5) | 0.06 | $t(\otimes)_{55}\epsilon[6,8]$ $\delta_{t55}{}^2 = 0.4$ | $c(\otimes)_{55}\epsilon[5,7]$ $\delta_{c55}{}^2 = 0.4$ | $q(\otimes)_{55}\epsilon[0.85,1]$ $\delta_{q55}{}^2 = 0.03$ | Normal |
| (5, 6) | 0.94 | $t(\otimes)_{56}\epsilon[36,44]$ $\delta_{t56}{}^2 = 1.6$ | $c(\otimes)_{56}\epsilon[26,34]$ $\delta_{c56}{}^2 = 1.6$ | $q(\otimes)_{56}\epsilon[0.9,1]$ $\delta_{q56}{}^2 = 0.02$ | Normal |
| Restrictions | | $T^{RG}\epsilon[66, 86]$ | $C^{RG}\epsilon[45, 64]$ | $Q^{RG}\epsilon[0.9, 1]$ | |

Easy to know: The earliest possible completion time is 66.4 days, when the cost is 636 thousand RMB and the relative quality of the product is 0.85. The latest possible completion time is 85.9 days, when the cost is 636 thousand RMB and the relative quality of the product is 0.85. The most likely completion time is 76.1 days, when the cost is a minimum of 452 thousand RMB and the quality of the product is 0.99.

**(2) Under contingency conditions**

The product now needs to be brought forward to 74 days and completed in around 70 days if possible, with a relative product quality of no less than 0.9 and the lowest possible total cost. So, how to redistribute the duration, cost and quality of the activities in the network in order to achieve the minimum cost and guarantee the quality of the product while keeping the duration required is the problem to be solved.

**a. Parameter selection and model building**

The weighting of the duration satisfaction function and the cost satisfaction function is 0.8 and 0.2 respectively, according to the expert scoring. The parameters in the cost-duration and quality-duration functions are assigned the following values: $\alpha_1 = \alpha_2 = 0.5$, indicating that the cost quadratic function is symmetric about $t_{ij} = \frac{t_{ij} + \overline{t_{ij}}}{2}$ and $\beta_1 = \beta_2 = 0.5$, indicating that the duration-quality function is symmetric about $t_{ij} = \frac{t_{ij} + \overline{t_{ij}}}{2}$ and $k_t = \frac{5}{2}$. Therefore, the duration-

cost-quality model for this aviation product for the contingency scenario are as follows:

$$
s.t \begin{cases}
max\ \mu \\
\mu = \omega_1\mu_1 + \omega_2\mu_2 \\
\mu_1 = \begin{cases} 1, & 66 \leq E(t) \leq 70 \\ \dfrac{74 - E(t)}{74 - 70}, & 70 \leq E(t) \leq 74 \\ 0, & else \end{cases} \\
\mu_2 = \begin{cases} 1, & E(c) \leq 45 \\ \dfrac{64 - E(c)}{64 - 45}, & 45 \leq E(c) \leq 64 \\ 0, & else \end{cases} \\
\underline{t_{ij}} \leq t_{ij} \leq \overline{t_{ij}} \\
\underline{q_{ij}} \leq q_{ij} \leq \overline{q_{ij}} \\
\underline{c_{ij}} \leq c_{ij} \leq \overline{c_{ij}} \\
66 \leq E(t) \leq 74 \\
0.9 \leq E(q) \leq 1 \\
E(c) \leq 64 \\
t^*_{ij} = 0.5\underline{t_{ij}} + 0.5\overline{t_{ij}} \\
q_{ij} = 1/1 + \left| \dfrac{t_{ij} - t^*_{ij}}{\overline{t_{ij}} - \underline{t_{ij}}} \right|^{5/2} \\
c_{ij} = k_{ij}\left(t_{ij} - t^*_{ij}\right)^2 + \underline{c_{ij}} \\
\omega_1 + \omega_2 = 1
\end{cases}
$$

Fitting the function of cost and duration to the data in Table 4 gives cost-duration functions as shown in Table 5.

### b. Model solving and results analysis

The model is solved using the adaptive variation particle swarm algorithm for finding the optimal solution. The parameters of the particle swarm algorithm were set: the population size

Table 5. Data Fitting for $k_{ij} - c_{ij}$.

| Activities | $k_{ij}$ | $c_{ij}$ |
|---|---|---|
| 1-->2 | 0.64 | $0.64(t_{12} - 7.5)^2 + 4$ |
| 1-->3 | 0.64 | $0.64(t_{13} - 7.5)^2 + 4$ |
| 1-->4 | 0.64 | $0.64(t_{14} - 7.5)^2 + 4$ |
| 2-->2 | 1.00 | $(t_{22} - 8)^2 + 6$ |
| 2-->5 | 0.80 | $0.8(t_{25} - 27.5)^2 + 15$ |
| 3-->3 | 0.75 | $0.75(t_{33} - 6)^2 + 3$ |
| 3-->5 | 0.40 | $0.4(t_{35} - 27)^2 + 12$ |
| 4-->5 | 1.33 | $1.33(t_{45} - 27.5)^2 + 16$ |
| 5-->5 | 2.00 | $2(t_{55} - 7)^2 + 5$ |
| 5-->6 | 0.50 | $0.5(t_{56} - 40)^2 + 26$ |

**Table 6. Results of model optimization.**

| Activities | Duration / day | Quality / 1 | Cost / ten thousand |
|:---:|:---:|:---:|:---:|
| 1-->2 | 5.6 | 0.920 | 6.28 |
| 1-->3 | 5.7 | 0.925 | 6.14 |
| 1-->4 | 5.8 | 0.936 | 5.88 |
| 2-->2 | 7.0 | 0.859 | 6.95 |
| 2-->5 | 26.0 | 0.957 | 16.69 |
| 3-->3 | 5.05 | 0.925 | 4.62 |
| 3-->5 | 23.03 | 0.951 | 15.73 |
| 4-->5 | 26.98 | 0.955 | 17.04 |
| 5-->5 | 6.73 | 0.937 | 5.92 |
| 5-->6 | 37.17 | 0.958 | 28.63 |
| $E(t) = 70.0$; $E(c) = 52.25$; $E(q) = 0.944$; $\mu = 0.924$ | | | |

was 100, the maximum number of iterations was 100, the inertia weight was 0.8, the self-learning factor was 0.5, the population learning factor was 0.5 and the self-variance probability was 0.15. Using MATLAB programming, the results were calculated as shown in Table 6.

It can be seen that when the main manufacturer controls each production link in the network according to the scheme in Table 6 in contingency situations, the supply time of the product can be advanced from 76.1 days to 70 days, at which time the most cost saving is about 522.5 thousand RMB, the relative quality of the product is 0.944 and the customer satisfaction can reach a maximum of 0.924.

### c. Analysis of weights

To further explore the changes in duration, cost and quality of this aviation product under contingency, a sensitivity analysis of the weights $\omega_1$ and $\omega_2$ in the model is now carried out and the results are shown in Table 7.

It can be seen that with different weights, the products can all be completed ahead of schedule to 70 days as required, at a cost of approximately 523 thousand RMB and a product quality of approximately 0.944. The comparison shows that the supply options for this aviation product do not differ significantly for different weighting values. It is important to note that the minimum duration weight is 0.51 because of the tighter schedule in emergency situations, and the maximum duration weight is 0.99 because economy is another factor to be considered.

### d. Suggested solutions for different levels of urgency

To explore the solution for this aviation product at different levels of urgency, the expected delivery time was used to represent the level of urgency, with a smaller expected delivery time

**Table 7. Comparison results for analysis of different weighting schemes.**

| | $\omega_1$ | $\omega_2$ | $E(t)$ | $E(c)$ | $E(q)$ | $\mu$ |
|:---:|:---:|:---:|:---:|:---:|:---:|:---:|
| 1 | 0.99 | 0.01 | 70.0 | 52.26 | 0.944 | 0.996 |
| 2 | 0.9 | 0.1 | 70.0 | 52.27 | 0.944 | 0.962 |
| 3 | 0.8 | 0.2 | 70.0 | 52.25 | 0.944 | 0.924 |
| 4 | 0.7 | 0.3 | 70.0 | 52.25 | 0.945 | 0.885 |
| 5 | 0.6 | 0.4 | 70.0 | 52.24 | 0.943 | 0.847 |
| 6 | 0.51 | 0.49 | 70.0 | 52.25 | 0.944 | 0.813 |

indicating greater urgency, taking $\omega_1 = 0.8$ and $\omega_2 = 0.2$, and the results obtained for the different scenarios are shown in Table 8.

It is known that the earliest completion time for the product to meet quality requirements is 68.1 days, at which point the cost is approximately 578 thousand RMB and the relative quality of the product is approximately 0.9. Therefore, the main manufacturer can use Table 7 as a reference to control each link of the supply chain network in order to obtain a reasonable solution for different schedule requirements in contingency situations.

Apparently, the model can well solve the problem of allocating the resource elements of each subject in a complex product supply chain network with restricted output results and multiple grey parameters under unexpected circumstances by transforming the multi-

**Table 8. Suggested solutions for different levels of urgency.**

| TRG | $E(t)$/ day | $E(c)$ / ten thousand | $E(q)$/ 1 | $\mu$ |
|---|---|---|---|---|
| 66 | 68.01 | 57.76 | 0.900 | 0.665 |
| $[t_{12}; t_{13}; t_{14}; t_{22}; t_{25}; t_{33}; t_{35}; t_{45}; t_{55}; t_{56}] = [5.5;5.5;5.4;7.0;25.4;4.1;22.9;26.3;6.1;36.7]$, $[c_{12}; c_{13}; c_{14}; c_{22}; c_{25}; c_{33}; c_{35}; c_{45}; c_{55}; c_{56}] = [6.68;6.65;6.69;6.93;18.37;5.68;18.88;17.99;6.68;31.50]$, $[q_{12}; q_{13}; q_{14}; q_{22}; q_{25}; q_{33}; q_{35}; q_{45}; q_{55}; q_{56}] = [0.903;0.904;0.903;0.862;0.902; 0.867; 0.900; 0.904; 0.876; 0.900]$ | | | | |
| 67 | 68.01 | 57.78 | 0.900 | 0.738 |
| $[t_{12}; t_{13}; t_{14}; t_{22}; t_{25}; t_{33}; t_{35}; t_{45}; t_{55}; t_{56}] = [5.5;5.5;5.4;8.9;25.4;4.0;22.9;26.3;6.1;36.7]$, $[c_{12}; c_{13}; c_{14}; c_{22}; c_{25}; c_{33}; c_{35}; c_{45}; c_{55}; c_{56}] = [6.63;6.69;6.72;6.88;18.45;5.90;18.86;17.95;6.53;31.51]$, $[q_{12}; q_{13}; q_{14}; q_{22}; q_{25}; q_{33}; q_{35}; q_{45}; q_{55}; q_{56}] = [0.905;0.903;0.902;0.869;0.900;0.855;0.901;0.906;0.888;0.900]$ | | | | |
| 68 | 68.01 | 57.79 | 0.900 | 0.865 |
| $[t_{12}; t_{13}; t_{14}; t_{22}; t_{25}; t_{33}; t_{35}; t_{45}; t_{55}; t_{56}] = [5.5;5.4;5.4;7.2;25.4;4.0;22.9;26.3;6.0;36.7]$, $[c_{12}; c_{13}; c_{14}; c_{22}; c_{25}; c_{33}; c_{35}; c_{45}; c_{55}; c_{56}] = [6.62;6.70;6.74;6.70;18.43;5.98;18.88;17.97;6.83;31.50]$, $[q_{12}; q_{13}; q_{14}; q_{22}; q_{25}; q_{33}; q_{35}; q_{45}; q_{55}; q_{56}] = [0.906;0.903;0.901;0.898;0.901;0.851;0.900;0.905;0.863;0.901]$ | | | | |
| 69 | 69.00 | 54.89 | 0.925 | 0.896 |
| $[t_{12}; t_{13}; t_{14}; t_{22}; t_{25}; t_{33}; t_{35}; t_{45}; t_{55}; t_{56}] = [5.5;5.8;5.5;7.0;25.8;4.2;23.7;26.7;6.0;37.0]$, $[c_{12}; c_{13}; c_{14}; c_{22}; c_{25}; c_{33}; c_{35}; c_{45}; c_{55}; c_{56}] = [6.56;5.77;6.45;6.92;17.39;5.35;16.40;16.94;6.82;30.58]$, $[q_{12}; q_{13}; q_{14}; q_{22}; q_{25}; q_{33}; q_{35}; q_{45}; q_{55}; q_{56}] = [0.908;0.940;0.913;0.863;0.934;0.885;0.940;0.960;0.864;0.919]$ | | | | |
| 70 | 70.0 | 52.25 | 0.944 | 0.924 |
| $[t_{12}; t_{13}; t_{14}; t_{22}; t_{25}; t_{33}; t_{35}; t_{45}; t_{55}; t_{56}] = [5.6;5.7;5.8;7.0;26.0;4.5;23.9;26.6;6.3;37.7]$, $[c_{12}; c_{13}; c_{14}; c_{22}; c_{25}; c_{33}; c_{35}; c_{45}; c_{55}; c_{56}] = [6.28;6.14;5.88;6.95;16.69;4.62;15.73;17.04;5.92;28.63]$, $[q_{12}; q_{13}; q_{14}; q_{22}; q_{25}; q_{33}; q_{35}; q_{45}; q_{55}; q_{56}] = [0.920;0.925;0.936;0.859;0.957;0.925;0.951;0.955;0.937;0.958]$ | | | | |
| 71 | 71.0 | 50.1 | 0.964 | 0.946 |
| $[t_{12}; t_{13}; t_{14}; t_{22}; t_{25}; t_{33}; t_{35}; t_{45}; t_{55}; t_{56}] = [5.9;6.1;6.0;7.1;26.3;4.9;24.6;26.9;6.4;38.0]$, $[c_{12}; c_{13}; c_{14}; c_{22}; c_{25}; c_{33}; c_{35}; c_{45}; c_{55}; c_{56}] = [5.66;5.23;5.51;6.74;16.22;3.95;14.35;16.46;5.65;28.01]$, $[q_{12}; q_{13}; q_{14}; q_{22}; q_{25}; q_{33}; q_{35}; q_{45}; q_{55}; q_{56}] = [0.945;0.961;0.950;0.892;0.971;0.960;0.972;0.983;0.958;0.970]$ | | | | |
| 72 | 72.0 | 48.4 | 0.978 | 0.965 |
| $[t_{12}; t_{13}; t_{14}; t_{22}; t_{25}; t_{33}; t_{35}; t_{45}; t_{55}; t_{56}] = [6.2;6.2;6.3;7.3;26.5;5.1;25.3;27.0;6.6;38.3]$, $[c_{12}; c_{13}; c_{14}; c_{22}; c_{25}; c_{33}; c_{35}; c_{45}; c_{55}; c_{56}] = [5.03;5.01;4.91;6.45;15.73;3.57;13.20;16.35;5.39;27.38]$, $[q_{12}; q_{13}; q_{14}; q_{22}; q_{25}; q_{33}; q_{35}; q_{45}; q_{55}; q_{56}] = [0.969;0.969;0.973;0.939;0.984;0.978;0.988;0.988;0.978;0.981]$ | | | | |
| 73 | 73.0 | 47.0 | 0.989 | 0.979 |
| $[t_{12}; t_{13}; t_{14}; t_{22}; t_{25}; t_{33}; t_{35}; t_{45}; t_{55}; t_{56}] = [6.5;6.6;6.6;7.4;26.7;5.2;25.4;27.0;6.6;38.9]$, $[c_{12}; c_{13}; c_{14}; c_{22}; c_{25}; c_{33}; c_{35}; c_{45}; c_{55}; c_{56}] = [4.64;4.48;4.54;6.35;15.52;3.49;12.99;16.32;5.27;26.60]$, $[q_{12}; q_{13}; q_{14}; q_{22}; q_{25}; q_{33}; q_{35}; q_{45}; q_{55}; q_{56}] = [0.983;0.988;0.986;0.954;0.990;0.982;0.990;0.990;0.986;0.993]$ | | | | |

objective problem into a single-objective non-linear programming problem, combined with the self-variant particle swarm algorithm, giving the exact values and reasonable solutions for each activity in the supply chain network and obtaining a satisfactory solution. In addition the weight sensitivity analysis shows that the weights do not significantly affect the results for this contingency for this aviation product. Therefore, the weight selection for this product allows the main manufacturer to reduce some of the resource input and further improve resource utilisation. By exploring the different levels of urgency, specific response options are obtained, which can be used as a basis for the main manufacturer when making decisions.

## 7. Conclusion

In this paper, a complex product supply chain network is modeled using an improved GERT network in solving realistic non-probabilistic uncertainty problems, considering the limitations of traditional project management tools such as PERT, CPM, etc. Combined with the demand for complex products in different emergency situations, the duration, product quality and cost of each supplier and manufacturer in the complex product supply chain network are studied, and the following conclusions are drawn:

1. By combining the historical data of the suppliers and the complex supply network, the main manufacturer can obtain reasonable optimised values for the duration, cost and quality of each supplier in the complex product supply network. This provides a basis for the main manufacturer's decision making and also provides ideas for the establishment and monitoring of contracts with regard to duration, cost and quality.

2. In contingency situations, a reduction in duration usually results in a corresponding change in cost and quality, although a satisfactory combination of duration, cost and quality can be optimised by suitable methods.

3. The weight values of certain options in different contexts and conditions may not necessarily have a large impact on the outcome, and when the option weight values have a limited impact on the outcome, the associated resource input can be reduced. This facilitates better resource utilisation. Sensitivity analysis is therefore important for programme decision-making.

The existing literature only explores revenue sharing, quality incentives, pricing strategies or elements related to a specific stage of the supply chain for complex products. In contrast to existing research, this paper presents a joint duration-cost-quality optimisation study of complex product supply chains in contingency situations, which is validated and discussed using data cases. And this paper is a further study that complements and enriches existing research. However, there are still certain limitations to this paper. For example, (1) the assumptions made in this paper on the relationship between duration-cost and duration-quality are quadratic and bell-shaped functions, which are not necessarily applicable to some specific processes. (2) The model requires a large amount of real historical data, and the authenticity of the data collected and obtained cannot be effectively guaranteed. (3) This paper simplifies the case study for ease of illustration, and certain specific processes in the case may be more complex and more conclusions may be drawn.

Based on the above research and conclusions, the following suggestions are made for the complex products supply chains:

a. Access to and sharing of data is a very important factor in complex product supply chains. Therefore, combining data sharing and blockchain technology with complex product supply chains will help drive complex product supply chains towards real-time feedback and

process control, and facilitate the development of complex product supply chain process management systems.

b. Increase research on process feedback in complex product supply chains, thereby furthering research on robustness and risk transfer in complex product supply chains.

## Author Contributions

**Conceptualization:** Peng Dong, Weimin Ye.

**Data curation:** Yunzhe Li.

**Formal analysis:** Peng Dong, Weimin Ye.

**Investigation:** Yunzhe Li, Weimin Ye.

**Methodology:** Yunzhe Li.

**Project administration:** Peng Dong, Weimin Ye.

**Resources:** Weimin Ye.

**Software:** Yunzhe Li.

**Supervision:** Peng Dong.

**Validation:** Yunzhe Li, Peng Dong.

**Writing – original draft:** Yunzhe Li, Peng Dong.

**Writing – review & editing:** Yunzhe Li, Peng Dong, Weimin Ye.

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
