## [Decision Letter · Decision Letter 0]

30 Jun 2023

PONE-D-23-15329Joint duration-cost-quality optimization model for complex product supply chains under contingency conditionsPLOS ONE

Dear Dr. Dong,

Thank you for submitting your manuscript to PLOS ONE. After careful consideration, we feel that it has merit but does not fully meet PLOS ONE’s publication criteria as it currently stands. Therefore, we invite you to submit a revised version of the manuscript that addresses the points raised during the review process.

We look forward to receiving your revised manuscript.

Kind regards,

Ibrahim Badi, PhD

Academic Editor

PLOS ONE

Journal Requirements:

Reviewers' comments:

Reviewer's Responses to Questions

**Comments to the Author**

1. Is the manuscript technically sound, and do the data support the conclusions?

Reviewer #1: Yes

Reviewer #2: Partly

2. Has the statistical analysis been performed appropriately and rigorously? 

Reviewer #1: I Don't Know

Reviewer #2: N/A

3. Have the authors made all data underlying the findings in their manuscript fully available?

Reviewer #1: Yes

Reviewer #2: No

4. Is the manuscript presented in an intelligible fashion and written in standard English?

Reviewer #1: Yes

Reviewer #2: Yes

5. Review Comments to the Author

Reviewer #1: Dear Author,

The paper presents the topic of model optimization for supply chains. Thank you for your submitted work. I am satisfied with the quality of the work, but I need additional information in order to reach a final conclusion.

Recommendations and comments for correction of work are as follows:

Comment 1:

Page 7 - It might be best if you numbered 3 objectives (

1. The first is to meet the requirements of shortening the duration, saving cost and guaranteeing quality of complex products under emergency situations, so as to respond to the delivery demand quickly and efficiently.

2. The second is to improve and enrich the theoretical research related to the supply chain of complex products, especially the content related to the supply chain of complex products in emergency situations.

3. The third is to provide decision basis for decision makers in the actual manufacturing of complex products based on the model findings.

Comment 2:

Page 7 – Error - The period at the end of the sentence is missing (mergency situationsThe third is to provide).

Comment 3:

Page 15- What are the currencies of the costs? Please indicate the currency in the entire text. (450 thousands to 640 thousands).

Comment 4:

My main comment and basic recommendation for improving the paper is to write the basic information in chapter five and to expand the basic input information about the case study. These data should be described in detail in terms of what they mean and what they are about. In the way it is now presented, I could not fully understand all the information.

Reviewer #2: Review comments

The manuscript proposed a joint duration-cost-quality optimization model for complex product supply chains under contingency conditions. Besides, a constructed model is applied to calculate specific arithmetic cases to give specific solutions for the supply network of this complex product.

The manuscript is interesting and proposes some implications for the integration of data sharing and blockchain technology with complex product supply chains. Accordingly, it can be reconsidered for publication after coping with the following issues:

(1) Please make each sentence within the abstract very short and comprehensive instead of lengthy sentence.

(2) Please, clearly state the motivations and contributions of your study.

(3) Please, make an extensive review of past studies related to complex product supply chains networks in terms methodology used, location, objective and compare them to your current study. A provision of summarized Table to highlight these studies will be very good presentation. Besides, it is requested that recent studies to be used as the ones used (references 17-20, 22, 23) within your study are very old and some dated back to 1987. References less than 5 years of publication are suggested (2018 to now).

(4) Please what is the advantage of the methodology applied in your study and why did you choose it rather than other existing methods.

(5) I suggested authors to numbering the equations within the manuscript as 1, 2, 3,.., etc.

(6) Please, uniform the size of writing style for Figures titles as well as Tables.

(7) The conclusion is too lengthy. I suggested authors to just highlight the main findings, the limitations of the study as well those of the method used. Also, authors should give some recommendations for future studies.

(8) The linguistic quality needs to be polished and several grammar mistakes need to be revised in the full context.

6. PLOS authors have the option to publish the peer review history of their article (what does this mean?). If published, this will include your full peer review and any attached files.

Reviewer #1: No

Reviewer #2: No

---

## [Author Response · Author response to Decision Letter 0]

13 Jul 2023

Dear Editors and reviewers:

We would like to take this opportunity to thank you for your professional work on our manuscript. Thank you for your great patience in waiting for this work to be completed.

We have made the careful modifications according to the reviewers’ constructive suggestions and submitted the‘Detailed Response to Reviewers’, hoping to meet with approval. In the revised manuscript (with changes marked), we used the red color to indicate the revised contents. We sincerely hope this manuscript will be finally acceptable to be published on PLOS ONE.

Once again, we would like to express our great appreciation to your consideration. We look forward to hearing from you regarding our resubmission. We would be glad to respond to any further questions and comments that you may have.

Sincerely yours, 

Dong

---

## [Decision Letter · Decision Letter 1]

15 Aug 2023

PONE-D-23-15329R1Joint duration-cost-quality optimization model for complex product supply chains under contingency conditionsPLOS ONE

Thank you for submitting your manuscript to PLOS ONE. After careful consideration, we feel that it has merit but does not fully meet PLOS ONE’s publication criteria as it currently stands. Therefore, we invite you to submit a revised version of the manuscript that addresses the points raised during the review process.

We look forward to receiving your revised manuscript.

Kind regards,

André Andrade Longaray, D.Sc.

Academic Editor

PLOS ONE

Journal Requirements:

Reviewers' comments:

Reviewer's Responses to Questions

**Comments to the Author**

1. If the authors have adequately addressed your comments raised in a previous round of review and you feel that this manuscript is now acceptable for publication, you may indicate that here to bypass the “Comments to the Author” section, enter your conflict of interest statement in the “Confidential to Editor” section, and submit your "Accept" recommendation.

Reviewer #1: (No Response)

Reviewer #2: All comments have been addressed

2. Is the manuscript technically sound, and do the data support the conclusions?

Reviewer #1: Yes

Reviewer #2: Yes

3. Has the statistical analysis been performed appropriately and rigorously? 

Reviewer #1: Yes

Reviewer #2: N/A

4. Have the authors made all data underlying the findings in their manuscript fully available?

Reviewer #1: (No Response)

Reviewer #2: Yes

5. Is the manuscript presented in an intelligible fashion and written in standard English?

Reviewer #1: Yes

Reviewer #2: Yes

6. Review Comments to the Author

Reviewer #1: Dear Author,

Thank you for submitting the corrected version of the paper! Thank you for submitting the corrected version of the paper. You made several mistakes in the additional literature review. When citing references, you should name the authors and you should not start the sentence with a lowercase letter. I will elaborate in the comments below.

Recommendations and comments for correction of work are as follows:

Comment 1:

A reference number cannot be the beginning of a sentence in this way. Write the last name of at least the first author.

Page 27- Section 2. Literature Review

1-“network topologies and analysed the effect of network topologies and network size on the bullwhip effect. [11] investigated the rules and characteristics of risk propagation in complex supply networks.”.

2- “opposite. [13] analysed complex manufacturing network systems in two dimensions, topological vulnerability and functional vulnerability, and gave recommendations for safe production.”.

3- “chain. [15] identified potential cascading failure problems in networks by studying the impact of initial events on the system causing maximum disruption consequences.”.

4- “[16] developed a multi-echelon supply chain evolution model based”.

5- “[20] proposed the concept”.

6- “of projects, [22] constructed”.

Comment 2:

Explain abbreviations when they are mentioned for the first time in the text.

“BA small-world networks or ER”

Comment 3:

These sentences seem unfinished to me. Please elaborate on these sentences.

Page 35–“ Where node 1 is a normalised virtual source node representing the layer 2 supplier. Nodes 2, 3 and 4 represent three layer 1 suppliers. Node 5 represents a main manufacturer, and node 6 represent a customer. In the supply chain network, node 1 “.

Reviewer #2: The authors have addressed all the comments. The manuscript is technically sound and presented in an intelligible fashion

7. PLOS authors have the option to publish the peer review history of their article (what does this mean?). If published, this will include your full peer review and any attached files.

Reviewer #1: No

Reviewer #2: No

---

## [Author Response · Author response to Decision Letter 1]

15 Aug 2023

We would like to take this opportunity to thank you for your professional work on our manuscript. Thank you for your great patience in waiting for this work to be completed.

We have made the careful modifications according to the reviewers’ constructive suggestions and submitted the‘Detailed Response to Reviewers’, hoping to meet with approval. In the revised manuscript (with changes marked), we used the blue color to indicate the revised contents. We sincerely hope this manuscript will be finally acceptable to be published on PLOS ONE.

Once again, we would like to express our great appreciation to your consideration. We look forward to hearing from you regarding our resubmission. We would be glad to respond to any further questions and comments that you may have.

---

## [Editor Report · Decision Letter 2]

11 Sep 2023

Joint duration-cost-quality optimization model for complex product supply chains under contingency conditions

PONE-D-23-15329R2

We’re pleased to inform you that your manuscript has been judged scientifically suitable for publication and will be formally accepted for publication once it meets all outstanding technical requirements.

Kind regards,

André Andrade Longaray, D.Sc.

Academic Editor

PLOS ONE
---

## [Editor Report · Acceptance letter]

2 Oct 2023

PONE-D-23-15329R2 

Joint duration-cost-quality optimization model for complex product supply chains under contingency conditions 

Dear Dr. Dong:

I'm pleased to inform you that your manuscript has been deemed suitable for publication in PLOS ONE. Congratulations! Your manuscript is now with our production department. 

Kind regards, 

on behalf of

Professor André Andrade Longaray 

Academic Editor

PLOS ONE